# Validation of the Arabic Version of the Edmonton Symptom Assessment System

**DOI:** 10.3390/ijerph20032571

**Published:** 2023-01-31

**Authors:** Omar Shamieh, Ghadeer Alarjeh, Mohammad Al Qadire, Zaid Amin, Abdelrahman AlHawamdeh, Mohammad Al-Omari, Omar Mohtadi, Abdullah Illeyyan, Omar Ayaad, Sawsan Al-Ajarmeh, Amal Al-Tabba, Khawlah Ammar, Dalia Al-Rimawi, Mahmoud Abu-Nasser, Fadi Abu Farsakh, David Hui

**Affiliations:** 1Department of Palliative Care, King Hussein Cancer Center, Amman 11941, Jordan; 2Center for Palliative & Cancer Care in Conflict, King Hussein Cancer Center, Amman 11941, Jordan; 3Faculty of Medicine, the University of Jordan, Amman 11942, Jordan; 4Adult Health Department, Faculty of Nursing, Al Al-Bayt University, Mafraq 25113, Jordan; 5College of Nursing, Sultan Qaboos University, Muscat 123, Oman; 6Department of Medicine, King Hussein Cancer Center, Amman 11941, Jordan; 7Sultan Qaboos Comprehensive Cancer Center Care and Research Center, Muscat P.O. Box 566 PC 123, Oman; 8Center of Research Shared Resources, King Hussein Cancer Center, Amman 11941, Jordan; 9MD Anderson Cancer Center, Houston, TX 77030, USA

**Keywords:** symptoms, ESAS, Arabic, psychometrics, palliative, cancer

## Abstract

Quality cancer care is a team effort. In addition, patients’ symptoms change over the course of treatment. As such, the Edmonton Symptom Assessment System (ESAS) is a simple tool designed to quickly monitor symptom change. Here, we present the results from a two-phase study aimed at validating the Arabic version of the ESAS (ESAS-A). Phase one involved the creation of two versions of the ESAS with both reverse and forward translations by bilingual, native Arabic speakers as well as evaluation by an expert panel. The reconciled version was then administered to 20 patients as a pilot from which to create the final version, which was then used with 244 patients. Phase two for the ESAS—involved an ESAS-based validation of 244 adults aged 18 years and older who were diagnosed with advanced cancer; then, further validation was completed in conjunction with two other symptom survey tools, the EORTC-Pal 15 and the HADS. The ESAS-A items possessed good internal consistency with an average Cronbach’s alpha of 0.84, ranging from 0.82 to 0.85. Moreover, the results of ESAS-A showed good agreement with those of EORTC QLQ- 15 PAL (r = 0.36 to 0.69) and HADS (r = 0.60 and 0.57) regarding anxiety and depression. We found the ESAS-A to be responsive to symptom change and a median time to completion of 3.73 min. The results of our study demonstrate that the ESAS-A is a reliable, valid, and feasible tool for the purposes of monitoring symptom change over the course of cancer treatment.

## 1. Introduction

Patients’ symptoms change over the course of treatment. Moreover, unlike in the early stages, advanced cancer is associated with additional physical and psychological symptoms as well as a lower health-related quality of life (HRQOL), which often result from the disease and treatment regimen [1,2,3,4]. However, the intensity of symptom burden is based on tumor burden; in addition, the toxicities and side effects of the treatment received, as well as the patient’s characteristics such as age, gender, and race also play a part [4,5,6]. Identifying the physical and psychological symptoms as well as their impact on patients’ QOL should be the focus of healthcare professionals in order to provide quality care [5,7]. As such, an adequate assessment of symptoms is a crucial prerequisite enabling healthcare professionals to reduce patients’ suffering, improve their quality of life, and ensure the effectiveness of patient care, especially within palliative care settings [3,6,8,9].

The assessment of such subjective experience needs to be underpinned by patients’ self-reporting. Therefore, symptoms are best evaluated by considering the patients’ expectancies, perceptions, values, beliefs, and culture [10,11]. Several assessment tools have been developed for the purpose of clinical research and practice. However, many received criticism for being lengthy, time-consuming, difficult to interpret, and not feasible in the context of clinical practice [12,13].

The Edmonton Symptom Assessment System (ESAS) was developed and validated to better assess the symptoms of cancer patients [14]. It is widely described throughout the literature and, moreover, has been translated into more than 20 languages worldwide, such as Spanish [12], Korean [15], Italian [16], Turkish [17], Japanese [18], and Portuguese [13,19]. The ESAS comprises 10 physical and psychological symptoms (i.e., pain, tiredness, drowsiness, nausea, lack of appetite, shortness of breath, depression, anxiety, feeling of well-being, and poor sleep) with scores ranging from 0 to 10 for each symptom. A score of 0 indicates an absence of the symptom, whereas a score of 10 indicates the worst possible severity. The time frame for all the items is the past 24 h [13]. Despite the original ESAS being widely used in Arabic-speaking countries for the purpose of research and clinical practice, it has not yet been validated in Arabic. The achievement of producing a valid, reliable, and sensitive Arabic version would facilitate its use in clinical trials as well as in clinical practices within Arabic-speaking countries [8,20]. On this note, we present the results from a two-phase study that aimed at validating the Arabic-translated version of the ESAS, which was conducted with patients who were diagnosed with advanced cancer in Jordan.

## 2. Materials and Methods

### 2.1. Study Design

This descriptive cross-sectional study consists of two phases. Phase 1: reverse translation and pilot testing was performed while using cognitive interviews for the purposes of semantic equivalence and cross-cultural adaptation of the Arabic ESAS version. Phase 2: psychometric testing of the Arabic-translated version of the ESAS in a large sample, which was conducted in conjunction with both the European Organization for Research and Treatment of Cancer quality of life questionnaire for palliative patients (EORTC QLQ C-15 PAL) and the hospital anxiety and depression scale (HADS) in order to confirm the psychometric properties of the scale in advanced cancer patients [6].

### 2.2. Participants

The inclusion criteria were: (1) patients with advanced cancer; (2) patients who are ≥18 years old; (3) patients who reported at least one symptom with a severity score of ≥ 4 out of 10; (4) patients with prior evaluation by the palliative care team as either inpatients or outpatients. Patients who possessed a cognitive impairment which was assessed by their clinicians using the diagnostic and statistical manual of mental disorders (4th ed.; DSM-IV) as well as patients who were too sick or refused to participate were all excluded.

### 2.3. Phase 1—Translation and Pilot Testing

Translation of the ESAS. The English version of the ESAS was reverse translated [21] for the purpose of this study. Accordingly, a forward translation of the English ESAS into Arabic was carried out by two independent, native Arabic bilingual-speaking professionals. Then, both Arabic translations were evaluated by a panel of five medical and nonmedical experts for clarity and cultural appropriateness. The panel reconciled the first Arabic version of the ESAS (i.e., the ESAS-A). Subsequently, two other independent bi-lingual nonmedical professionals translated the reconciled ESAS-A back into English. The expert panel examined the reconciled English version against the original ESAS. After thoroughly reviewing all forward and backward translations, including the reconciled versions, the expert panel produced the second improved version of the ESAS-A.

Pilot testing of the ESAS-A. The provisional version was administered to 20 patients to pre-test the ESAS-A with a focus on evaluating its clarity, acceptability, and cultural appropriateness. Patients were asked if any items were difficult to answer or understand, or whether they were confusing, upsetting, or required in the form of alternative wording.

### 2.4. Phase 2—Psychometric Evaluation

From the period of May 2015 to May 2017, two research assistants identified eligible participants from the inpatient and outpatient medical records. All recruited participants signed a consent form and were interviewed face-to-face in order to fill out all of the required forms.

*Demographic and Clinical Data*. The demographic data were recorded during the interview and included: date of birth, gender, nationality, religion, level of education, and place of residence. The clinical information obtained from the medical records were the primary disease, date of diagnosis, treatment received, and Karnofsky performance status (KPS).

*Questionnaires.* The participants were then requested to complete the EASA-A, QLQ-C15-PAL, and HADS. In addition to those, the time taken for patients to complete the questionnaires and their opinions regarding the ESAS-A were recorded. Furthermore, a group of patients was asked to complete the three questionnaires a second time (2 to 3 weeks from baseline) for the purpose of conducting response to change analyses (RCA).

1.*The Edmonton Symptom Assessment Scale (ESAS)* is a self-reported numeric rating scale; furthermore, it is used to evaluate the most common physical and psychological symptoms in the context of palliative settings. The ESAS comprises 10 physical and psychological symptoms (i.e., pain, tiredness, drowsiness, nausea, lack of appetite, shortness of breath, depression, anxiety, feeling of well-being, and poor sleep) with scores ranging from 0–10 for each symptom. A score of 0 indicates an absence of the symptom, whereas a score of 10 indicates the worst possible severity. The time frame for all symptoms is the past 24 h. [13]. The original English version of the ESAS showed that the scale is valid and reliable, whereby the overall Cronbach’s alpha for the ESAS instrument was 0.79 [22]2.*EORTC QLQ C—15 PAL*. The EORTC QLQ C—15 PAL system comprises 14 items that are grouped into two multi-item subscales, including: functional scales, symptom scales, and a single item assessing general health status (GHS). Patients rate the 14 symptoms using a four-point Likert response scale with the labels of: 1—not at all; 2—a little; 3—quite a bit; and 4—very much. Further, a seven points Likert response scale, ranging from 1 (very poor) to 7 (excellent) was utilized for the GHS item. No time frame is specified for the physical functioning scale items. The time frame for all remaining items is the past week. The Arabic version has been validated; further, in respect of this, Cronbach’s alpha for the questionnaire was 0.7. Moreover, confirmatory factor analysis was deemed to meet the goodness-of-fit criteria. In addition, the convergent validity was measured, and all items exceeded 0.40, thereby indicating satisfactory convergent validity [23].3.*Hospital Anxiety and Depression Scale (HADS)*. The HADS tool was developed in 1983 by Zigmond and Snaith. Since then, it has been validated and used in order to assess non-psychiatric patients for anxiety and depression [24]. Further, HADS possessed two subscales assessing anxiety and depression (7 items for each subscale) on a 4-point Likert scale [25]. For each sub-scale, the total score ranges from 0 to 21, where a score of 8 or above is considered abnormal [24,25]. Furthermore, the Arabic version of the HADS is confirmed to be valid and reliable [26].

### 2.5. Data Analysis

The sample size of this study was estimated based on recommendations by psychometric analysis experts and previous ESAS validation studies, which indicate a sample size ranging between 34 and 241 is appropriate [12,27]. A minimum sample size of 200 participants was required for a statistically significant difference of 0.2, an alpha error of 5%, two-tailed testing, and a power of 80%.

Descriptive statistics (i.e., means, standard deviations, frequencies, and percentages) were utilized for summarizing the patients’ demographic and clinical data. Internal consistency was assessed by calculating Cronbach’s alpha. The measurement was considered reliable if Cronbach’s alpha >0.7 [21].

Criterion validity was assessed by measuring the correlation between the item scores on the ESAS-A and its corresponding score on the Arabic EORTC quality of life questionnaire and the Arabic HADS on the basis of Pearson’s correlation coefficient. Pearson’s correlations of 0–0.19, 0.2–0.39, 0.4–0.59, 0.6–0.79, and 0.8–1.0; correspond to very weak, weak, moderate, strong, and very strong, respectively [28].

The known group validity was used to evaluate the extent to which the ESAS-A could detect the difference in the treatment setting between the inpatient, outpatient, PPS, and gender. Analysis by one-way ANOVA and comparison of means between the inpatient and outpatient settings, and the females and males were conducted. A PPS of > or ≤70%and a *p*-value of <0.05 were considered.

In order to test for responsiveness to change, we asked 53 outpatients to repeat the ESAS-A, the EORTC-QLQ-15-PAL, and the HADS after 2–3 weeks of their return to palliative care. Responsiveness was determined by comparing the change in ESAS-A item scores with the change in the Arabic version of the EORTC QLQ 15 PAL or HADS (for anxiety and depression) scores at both the baseline and after 2–3 weeks. Anxiety and depression were not included in the EORTC QLQ 15 PAL; as such, the Arabic HADS was used instead. Pearson’s correlation was used to measure the correlation between score changes in each item in all instruments. Furthermore, a *p*-value of <0.05 was considered significant.

Finally, we measured the completion time of the ESAS-A, and the Statistical Package for the Social Sciences (SPSS) Version 19 was used for the purpose of data analysis.

## 3. Results

### 3.1. Phase 1—Translation/Pilot Testing

The final translated version of the ESAS-A was approved by the expert panel. Regarding the pilot test, a total of 20 patients completed the cognitive interviews for the final Arabic version. The pilot results showed that certain patients did not clearly understand the words fatigue, drowsiness, well-being, and/or sleep. Patients’ comments were reviewed, and then the committee decided to add two Arabic words explaining the word fatigue (tiredness or exhaustion) and two words for drowsiness (sleepy or lethargic). The expert panel used semantic equivalence for this purpose. Well-being was replaced with a sentence explaining the meaning of well-being as feeling healthy and active. After the final review, the committee agreed on the final version of the ESAS-A, which is the version that was used in this study.

### 3.2. Phase 2—Validation Psychometric Properties

#### 3.2.1. Patient Demographics

A total of 294 eligible patients were approached, of these, there were 163 outpatients and 128 inpatients. In addition, there were 27 patients from outpatient settings (16.6%). Further, there were certain outpatients who declined to participate due to lack of interest, while 20 (15.6%) inpatients were too ill. Moreover, 3 patients did not complete the study. As such, the final sample was 244: 108 (44.2%) inpatients and 136 (56%) outpatients. In regards to 50% (*n* = 122) of the females, the mean age was 52.8 (with a 14.6 range of years). In addition, most of the sample, 75.4% (*n* = 184), were married and held a secondary school level of education, 38.8% (*n* = 93). Furthermore, 82.7% (*n* = 202) of the participants were not working. The mean PPS was 58.4. The complete participants’ demographics and clinical characteristics are presented in Table 1.

#### 3.2.2. Internal Consistency

Table 2 presents the results of internal consistency for the ESAS-A. The results show that all the ESAS-A items possess a good internal consistency with an overall Cronbach’s alpha of 0.84. However, no significant change was observed in the value for Cronbach’s alpha when any of the symptoms were deleted. Furthermore, the value of Cronbach’s alpha ranged from 0.82 to 0.85. Moreover, the prevalence of symptoms ranged from 45% to 86.7%. The most reported symptom was tiredness (86.7%), followed by pain (83.3%). In contrast, the least reported symptoms were nausea (37.1%) and depression (45.0%).

#### 3.2.3. Criterion Validity

The results show that all the ESAS-A items are significantly correlated with similar symptoms in the Arabic version of the EORTC QLQ- C15 PAL. The strongest correlation was found between the loss of appetite as recorded in the ESAS-A and the EORTC QLQ-C15 PAL (r = 0.69), followed by nausea in the two instruments (r = 0.65). Table 3 presents the correlation coefficients between the items of the ESAS-A and similar items on the EORTC QLQ 15 PAL. Anxiety and depression were not included in the EORTC QLQ-C15 PAL; as such, the Arabic HADS was utilized instead. In addition, the anxiety and depression scores in the ESAS-A and the Arabic HADS were significantly correlated (r = 0.60 and 0.57), respectively.

#### 3.2.4. Known Group Validity

Our results showed a higher mean for all the ESAS symptoms in the inpatients when compared to the outpatients. Similarly, the mean for all the ESAS symptoms was higher for patients with a lower PPS than those with higher PPS scores. Furthermore, the mean for all the ESAS symptoms was higher in females when compared to males. We found statistically significant differences between females and males in 3 of the ESAS-A symptoms, where females suffer more from depression (4.22 vs. 2.61, *p* ˂ 0.001), anxiety (4.98 vs. 3.51, *p* = 0.001), and shortness of breath (3.82 vs. 2.84, *p* = 0.021), as shown in Table 4.

#### 3.2.5. Responsiveness to Change

The correlation between score changes IN THE symptoms in all instruments showed significant results except for drowsiness/fatigue, which showed no significant result (*p =* 0.068). However, the strongest correlation (−0.647, *p* ˂ 0.0001) was found between well-being and physical function. (Table 5)

#### 3.2.6. Completion Time

The time required to fill the ESAS-A was evaluated by measuring the mean completion time for the total sample of 244 patients. The median time was 3.73 min and the IQR was 2.42 and 5.7 min.

## 4. Discussion

This study demonstrates the reliability and validity of the ESAS-A for use in Arabic-speaking patients with advanced cancer who are receiving palliative care in Jordan. In this study, the internal consistency coefficient is 0.84. Indeed, this value is considered one of the highest reported values when considering previous validation studies in which Cronbach’s alpha ranged from 0.71 to 0.88 [12,15,18,29,30,31]. Therefore, we find our result is consistent with those other studies. Moreover, it can be said that the ESAS-A possesses good internal consistency. Criterion validity is an effective measure for determining the actual validity of a test score. Our results show moderate to strong correlations between all the ESAS-A items as well as with similar items in the Arabic version of the EORTC QLQ-C15 PAL and the HADS. While the appetite item was found to be the strongest correlation between the two questionnaires (r = 0.682), the weakest correlations were found between the well-being item in the ESAS-A and the global health item in the EORTC QLQ 15 PAL (r = −0.366). Our results are consistent with the results of the Croatian study for appetite, pain, and shortness of breath, and three is moderate to strong correlation when comparing these items between the two questionnaires (r = −0.649, 0.754, and −0.718, respectively). In addition, our results are inconsistent with the Croatian study in regard to the ESAS item regarding wellbeing (r = −0.642 with EORTC QLQ-C15 PAL global). These differences may be partly explained by differences in patient characteristics [32].

The same goes for the correlation between items of anxiety and depression in the ESAS-A and the emotional function in QLQ- C15 PAL, which showed moderate and high correlation (r = −0.676 and −0.585), respectively. We attribute this to a phenomenon that has already been demonstrated by the medical community, i.e., that anxiety and depression may affect emotional functioning. However, our results are also consistent with the Croatian study in that they show a strong correlation between depression and anxiety in the ESAS and the emotional function in the QLQ-C15 PAL (−0.711 and −0.773), respectively. Accordingly, it can be assumed that the ESAS-A can predict the QLQ-C15 PAL results when measuring the same construct. On the other hand, the ESAS-A shows a strong and moderate correlation considering the two items of anxiety and depression (r = 0.601 and −0.571, respectively). Indeed, this is quite similar to the results of the study by Pavia and colleagues [19].

Overall, our results showed that 100% of the ESAS-A items correlated well with similar items in the QLQ-C15 PAL (>0.40). Accordingly, it can be assumed that the ESAS-A and QLQ-C- 15 PAL measure the same construct.

The known-group validity in this study shows that inpatients possess more severe symptoms than outpatients, which is expected due to the fact that inpatients tend to have more distressing symptoms than outpatients. However, such results contradict those in a previous study where palliative inpatient and outpatient settings showed similarities in physical and psychological symptom severity as well as increased pain intensity in outpatient settings [33].

Moreover, our results show that patients with a lower PPS have higher symptom severity than those with a higher PPS; this may be because all of the patients enrolled in this study had advanced cancer. Moreover, the PPS of this group of patients is found to usually decrease in conjunction with the increase in the severity of the physical and psychological symptoms that result from the disease and the treatment. These results correspond with those of the study by Paiva and colleagues, which showed higher symptom scores for patients with lower PPS scores [19].

Furthermore, our study showed that females had higher levels of anxiety, depression, and shortness of breath compared to males. These results correspond with the previous studies where females tend to develop anxiety disorders and depression more than males during their illness and life trajectory [34,35,36,37].

A responsiveness test has been used in many studies on ESAS validation [13,38,39]. In our study, the responsiveness test demonstrated significant results, with moderate-to-strong correlations for all symptom changes, except for drowsiness, where the correlation was weak and not significant. This could be related to the fact that drowsiness and fatigue are not similar items. Moreover, there is no similar item in relation to drowsiness in the EORTC QLQ 15 PAL tool to which correlation can be assessed. As such, the results indicate that the ESAS-A could be used to differentiate and measure the status of patients, both before and after receiving palliative care services.

Finally, the completion time was evaluated in this study by identifying the needed time in which to fill the scale. Around 3.73 min were needed to complete the ESAS-A, which is considered within the range of the other studies that were mentioned (i.e., 2–7.2 min) [12,40,41,42]. Moreover, there was no significant difference in the time required to complete the scale between the ESAS-A, the Arabic version of the HADS, and the Arabic version of the EORTC QLQ 15 PAL. Based on this, it can be concluded that the ESAS-A is feasible for use in patients with advanced cancer and who are receiving palliative care. Future studies can target diverse patient populations, including more patients with early-stage cancer and non-cancer patients.

The limitations of our study are as follows: (1) This study was conducted in a tertiary care cancer center, which may lead to selection bias and could affect the generalizability of our study conclusion; (2) Only patients with advanced cancer were included in this study; (3) a majority (96%) of study participants were Jordanians. We do not expect major differences in the ESAS-A interpretation when applied in other Arab countries; however, further testing is required in order to support this notion. (4) We did not assess test–retest reliability. (5) Responsiveness was only tested in the outpatient setting and thus these findings may not apply to hospitalized patients who are, typically, acutely ill. Future studies may be required to examine the ESAS in more diverse patient populations including those with non-cancer diseases.

## 5. Conclusions

Our results demonstrate that the Arabic ESAS (i.e., the ESAS-A) is responsive to change and is a valid, reliable, and feasible tool for symptom assessment in palliative cancer settings. Furthermore, the ESAS-A can be routinely used to assess the symptoms of patients with advanced cancer in Jordan and other Arabic-speaking countries.

## Figures and Tables

**Table 1 ijerph-20-02571-t001:** Patient Demographics and Clinical Characteristics (*n* = 244).

Variables	Frequency (%)
** *Age Mean (SD)* **	52.8 (14.6%)
*Gender*	
Female	122 (50%)
Male	122 (50%)
*Nationality*	
Jordanian	234 (95.9%)
Non-Jordanian	10 (4.1%)
*Marital status*	
Single	38 (15.6%)
Married	184 (75.4%
Divorced	1 (0.4%)
Widowed	21 (8.6%)
*Education*	
Primary school or lower	48 (19.6%)
Secondary school	93 (38.8%)
Diploma	29 (12.1%)
Bachelor’s	62 (25.8%)
Postgraduate (Master or PhD)	12 (5%)
*Working status*	
Working	42 (17.2%)
Not working	202 (82.7%)
*Governorates*	
Amman	168 (69.1%)
Zarqa’a	28 (11.5%)
Irbid	18 (7.4%)
Others	30 (11.2%)
*Clinical setting*	
Inpatient	108 (44.2%)
Outpatient	136 (56%)
*Unit*	
Leukemia	3 (1.2%)
Medical	103 (42.2%)
Palliative	130 (53.5%)
Surgical	8 (3.3%)
*Cancer Site*	
Breast	50 (20.5%)
Head and neck	12 (4.9%)
Lung cancer	19 (7.8%)
Brain cancer	3 (1.2%)
Gastrointestinal	64 (26.2%)
Skin and soft tissue	11 (4.5%)
Gynecological	19 (7.8%)
Hematologic	29 (11.9%)
Genitourinary	26 (10.7%)
Others	11 (4.5%)
*PPS and Mean (SD)*	58.4 (21.5)

PPS: palliative performance scale and SD: standard deviation.

**Table 2 ijerph-20-02571-t002:** Internal Consistency of the ESAS-A.

Symptoms ESAS-A	Symptom Prevalence *n* (%)	ESAS-A Mean Score (SD)	Cronbach’s Alpha If Item Deleted	Cronbach’s Alpha
Pain	200 (83.3%)	6.97 (1.860)	0.84	
Fatigue	208 (86.7%)	7.30 (1.877)	0.82	
Nausea	89 (37.1%)	6.63 (2.002)	0.85	
Depression	108 (45.0%)	6.96 (1.995)	0.82	0.843
Anxiety	131 (54.6%)	7.15 (1.981)	0.83	
Drowsiness	177 (73.8%)	7.12 (1.906)	0.82	
Appetite	174 (72.5%)	7.40 (2.177)	0.82	
Well-being	193 (80.4%)	7.38 (1.876)	0.82	
Shortness of breath	111 (46.3%)	6.56 (1.813)	0.83	
Sleepiness	169 (70.4%)	6.85 (1.882)	0.83	

**Table 3 ijerph-20-02571-t003:** Criterion Validity (*n* = 244).

Symptoms ESAS-A	Instrument	Item	Correlation Coefficient (95%CI)	Sig	Level
Pain	A-EORTC QLQ-C15	Pain	0.490 (0.378–0.598)	˂0.0001	Moderate
Fatigue	A-EORTC QLQ-C15	Fatigue	0.522 (0.413–0.628)	˂0.0001	Moderate
Nausea	A-EORTC QLQ-C15	Nausea and vomiting	0.657 (0.562–0.754)	˂0.0001	Strong
Depression	A-HADS	Depression	0.571 (0.467–0.675)	˂0.0001	Moderate
	A-EORTC QLQ-C15	Emotional function	−0.676 (−0.771–−0.583)	˂0.0001	Strong
Anxiety	A-HADS	Anxiety	0.601 (0.500–0.703)	˂0.0001	Strong
	A-EORTC QLQ-C15	Emotional function	−0.585 (−0.691–−0.484)	˂0.0001	Moderate
Appetite	A-EORTC QLQ-C15	Loss of appetite	0.692 (0.601–0.785)	˂0.0001	Strong
Well-being	A-EORTC QLQ-C15	Global health	−0.366 (−0.483–−0.246)	˂0.0001	Weak
	A-EORTC QLQ-C15	Physical function	−0.573 (−0.676–−0.468)	˂0.0001	Moderate
Shortness of breath	A-EORTC QLQ-C15	Dyspnea	0.618 (0.520–0.720)	˂0.0001	Strong
Poor sleep	A-EORTC QLQ-C15	Sleep disturbance	0.633(0.534–0.731)	˂0.0001	Strong
Drowsiness	A-EORTC QLQ-C15	Physical function	−0.511 (−0.618–−0.401)	˂0.0001	Moderate
	A-EORTC QLQ-C15	Fatigue	0.469 (0385–0.604)	˂0.0001	Moderate
TSDS	A-EORTC QLQ-C15	Global health	−0.488 (−0.600–−0.376)	˂0.0001	Moderate

ESAS-A: Arabic version of the Edmonton Symptom Assessment System; QLQ-C15PAL: European Organization for Research and Treatment of Cancer quality of life questionnaire for palliative patients; HADS: hospital anxiety and depression scale; TSDS: total symptom distress score.

**Table 4 ijerph-20-02571-t004:** Known Group Validation (*n* = 244).

ESAS-A	Care Setting		*p*-Value	PPS		*p*-Value	Gender		*p*-Value
	Inpatient (*n* = 53)	Outpatient (*n* = 191)		≥70% (*n* = 103)	<70% (*n* = 141)		Female (*n* = 122)	Male (*n* = 122)	
	Mean (SD)	Mean (SD)		Mean (SD)	Mean (SD)		Mean (SD)	Mean (SD)	
Pain	6.32 (2.73)	5.90 (2.72)	0.316	5.41 (2.79)	6.47 (2.62)	0.003	6.01 (2.83)	5.97 (2.63)	0.907
Fatigue/Tiredness	7.92 (1.96)	6.00 (2.83)	˂0.001	5.04 (2.82)	7.48 (2.26)	˂0.001	2.66 (3.26)	2.81 (3.31)	0.726
Nausea	3.70 (3.34)	2.47 (3.21)	0.016	1.83 (2.98)	3.44 (3.35)	˂0.001	6.55 (2.86)	6.29 (2.69)	0.462
Depression	4.74 (3.24)	3.05 (3.52)	0.002	2.50 (3.47)	4.19 (3.41)	0.001	4.22 (3.73)	2.61 (3.11)	˂0.001
Anxiety	6.02 (2.99)	3.76 (3.55)	˂0.001	3.48 (3.59)	4.96 (3.40)	0.002	4.98 (3.64)	3.51 (3.32)	0.001
Drowsiness	7.25 (2.56)	4.97 (3.21)	˂0.001	4.08 (3.09)	6.57 (2.87)	˂0.001	5.66 (3.40)	5.27 (3.02)	0.341
Appetite	7.81 (2.71)	4.98 (3.39)	˂0.001	3.96 (3.12)	6.93 (3.12)	˂0.001	5.32 (3.55)	5.87 (3.34)	0.215
Well-being	8.35 (1.95)	5.58 (2.96)	˂0.001	4.64 (3.08)	7.43 (2.24)	˂0.001	6.28 (3.12)	6.05 (2.89)	0.559
Shortness of breath	4.79 (3.36)	2.93 (3.16)	˂0.001	2.27 (2.84)	4.19 (3.40)	˂0.001	3.82 (3.45)	2.84 (3.08)	0.021
Poor sleep	6.42 (2.69)	4.71 (3.20)	˂0.001	4.13 (3.23)	5.92 (2.90)	˂0.001	5.47 (3.22)	4.70 (3.10)	0.057
TSDS	63.35 (12.8)	44.52 (20.6)	˂0.001	37.32 (19.8)	57.83 (16.4)	˂0.001	51.12 (20.83)	45.97 (20.21)	0.053

SD, Standard Deviation; TSDS, Total symptom distress score.

**Table 5 ijerph-20-02571-t005:** Responsiveness of the ESAS-A for outpatients (*n* = 53).

ESAS-A	Pearson Correlation Coefficient (95% CI)		*p*-Value
	A-EORTC QLQ-C15—Pal	A-HADS	
Pain	0.561 (0.342–0.854)		˂0.0001
Fatigue/tiredness	0.567 (0.293–0.721)		˂0.0001
Nausea	0.636 (0.389–0.811)		˂0.0001
Depression/HADS		0.456 (0.232–0.808)	0.001
Anxiety/HADS		0.529 (0.315–0.841)	˂0.0001
Appetite	0.597 (0.332–0.757)		˂0.0001
shortness of breath	0.465 (0.197–0.685)		0.001
Poor sleep	0.623 (0.369–0.800)		˂0.0001
Well-being/ physical function	−0.647 (−0.832–−0.408)		˂0.0001
Drowsiness/ sleeping	0.261 (−0.018–0.483)		0.068

ESAS-A: Arabic version of the Edmonton Symptom Assessment System; A-EORTC—Pal-15: Arabic version of the EORTC Palliative 15; and A-HADS, Arabic version of the hospital anxiety and depression scale.

## Data Availability

Not applicable.

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
