# Peer review of "Validation of the Arabic Version of the Edmonton Symptom Assessment System"

_ijerph, 2023, doi:10.3390/ijerph20032571_

Round 1

Reviewer 1 Report

The abstract is difficult to follow and needs extensive editing. Here is a proposed sequence for the abstract: 

Quality cancer care is a team effort. Patient symptoms change over the course of treatment.  The Edmonton Symptom Assessment System (ESAS) is a simple tool designed to quickly monitor symptom change. This report presents the results from two-part study whose goal is validation of an Arabic version.  244 adults aged 18 years and older with advanced cancer participated in the study.  Part one of this process involved creation of two versions of ESAS with reverse and forward translation by bilingual, native Arabic speakers, and evaluation by an expert panel.  The reconciled version was then administered to 20 patients to create the final version for use with 244 patients. For comparison, these patients also completed two other symptom survey tools. The ESAS-A is responsive to symptom change with a median time to completion of 3.73 minutes. Psychometric comparison with two other assessment tools provides evidence that ESAS-A is a valid, reliable, and feasible tool for monitoring symptom change over the course of cancer treatment.  

Overall, this paper might be improved by breaking it into two papers. 

This is a part of the presentation that could be its own paper: 

The results showed that some patients did not clearly understand the words fatigue,  drowsiness, well-being, and sleep. Patients' comments were reviewed, and then the committee decided to add two Arabic words explaining the word fatigue (tiredness or exhaustion) and two words for drowsiness (sleepy or lethargic). The expert panel used semantic equivalence. Well-being was replaced with a sentence explaining the meaning of well being as feeling healthy and active. We added an explanation for "Best Sleep" (no problems with sleep) based on patients' and expert panel recommendations. After the final review, the committee agreed on the final version of ESAS-A (Appendix – A) used in this study.

These symptoms are common to all. Having a deeper understanding of the interpretation of these symptoms and the linguistic approach their expression would assist practitioners world wide who are treating Arabic speakers.   

Author Response

Validation of the Arabic Version of the Edmonton Symptom Assessment System

Reviewers Comments

Reply to the Reviewers Comments

Reviwer1

Extensive editing of English language and style required

Thank you for your comment:

Based on your feedback, we have sent the manuscript for scientific editing through the publisher. We did not receive the final edited version yet.  However, we are returning this reply first before the final reviewed manuscript to prevent further delay. 

The abstract is difficult to follow and needs extensive editing. Here is a proposed sequence for the abstract:

Quality cancer care is a team effort. Patient symptoms change over the course of treatment.  The Edmonton Symptom Assessment System (ESAS) is a simple tool designed to quickly monitor symptom change. This report presents the results from two-part study whose goal is validation of an Arabic version.  244 adults aged 18 years and older with advanced cancer participated in the study.  Part one of this process involved creation of two versions of ESAS with reverse and forward translation by bilingual, native Arabic speakers, and evaluation by an expert panel.  The reconciled version was then administered to 20 patients to create the final version for use with 244 patients. For comparison, these patients also completed two other symptom survey tools. The ESAS-A is responsive to symptom change with a median time to completion of 3.73 minutes. Psychometric comparison with two other assessment tools provides evidence that ESAS-A is a valid, reliable, and feasible tool for monitoring symptom change over the course of cancer treatment. 

 Thank you for the worthwhile suggestion:

The abstract was modified according to your suggestion

 The abstract reads as follows

Quality cancer care is a team effort. Patients’ symptoms change over the course of treatment. The Edmonton Symptom Assessment System (ESAS) is a simple tool designed to quickly monitor symptom change. This study presents the results from two-phase study aimed to validate the Arabic version of ESAS (ESAS-A Phase one involved creation of two versions of ESAS with reverse and forward translation by bilingual, native Arabic speakers, and evaluation by an expert panel. The reconciled version was then administered to 20 patients as piloting to create the final version for use with 244 patients. Phase two for ESAS – A validation; 244 adults aged 18 years and older with advanced cancer completed ESAS – A in conjunction with two other symptom survey tools (EORTC-Pal 15) and HADS. ESAS-A items had good internal consistency, Cronbach's Alpha is 0.84 ranged from 0.82 to 0.85. ESAS-A was also associated with EORTC QLQ- C15 PAL (r=0.36 to 0.69) and HADS anxiety and depression (r=0.60 and 0.57). ESAS-A was responsive to symptom change and the median time to completion was 3.73 minutes. Our study found that ESAS-A is a reliable, valid, and feasible tool for monitoring symptom change over the course of cancer treatment.

Overall, this paper might be improved by breaking it into two papers.

This is a part of the presentation that could be its own paper:

The results showed that some patients did not clearly understand the words fatigue, drowsiness, well-being, and sleep. Patients' comments were reviewed, and then the committee decided to add two Arabic words explaining the word fatigue (tiredness or exhaustion) and two words for drowsiness (sleepy or lethargic). The expert panel used semantic equivalence. Well-being was replaced with a sentence explaining the meaning of well-being as feeling healthy and active. We added an explanation for "Best Sleep" (no problems with sleep) based on patients' and expert panel recommendations. After the final review, the committee agreed on the final version of ESAS-A (Appendix – A) used in this study.

These symptoms are common to all. Having a deeper understanding of the interpretation of these symptoms and the linguistic approach their expression would assist practitioners worldwide who are treating Arabic speakers.  

Your suggestion is highly appreciated.   We discussed among our research team and feel that it would be best at this stage to present all findings in one paper so readers can understand both the derivation and the validation of the ESAS-A in one paper.  However, we added more clarification on the methodology and split it into 2 phases as you suggested.  Nevertheless, we will leave it up to the editor to decide if we should remove this information and consider publishing it in another manuscript.

Reviewer 2 Report

the paper is on a very interesting topic. It address a specific need: the need of translation of psychometric instruments in different languages. The paper is well designed and well written. I suggest only some minor changes: first, I suggest to better describe results according to previous papers; second, I suggest to discuss gender differences, if any, third, I suggest to discuss in a deeper way any limitations of the study. 

Author Response

Validation of the Arabic Version of the Edmonton Symptom Assessment System

Reviewers Comments

Reply to the Reviewers Comments

Reviwer2

the paper is on a very interesting topic. It addresses a specific need: the need of translation of psychometric instruments in different languages. The paper is well designed and well written. I suggest only some minor changes:

First, I suggest to better describe results according to previous papers

Many thanks for the suggestions:

We have now modified the discussion based on the Croatian and Brazilian studies as follows:

1st: presented the translation process and pilot-testing of the ESAS-A

2nd: presented the validation and psychometric properties of the updated version of ESAS-A as follow;

-   Patient demographics & Clinical Characteristics

-   Internal Consistency of the ESAS-A items

-   Correlation between the ESAS-A and the other validated tools

-   Known group comparisons (care setting, KPS and gender)

-    Responsiveness-to-change analysis (RCA)

-   Finally: presented the completion time  of the ESAS-A

Second, I suggest to discuss gender differences, if any

Thank you for the crucial comment:

After analyzing the at Females Vs Males differences in regard to ESAS-A scores

We found significant differences between females vs males in the ESAS-A items: depression (4.22 Vs 2.61, P=.000), anxiety (4.98 Vs 3.51, P=.001), and shortness of breath (3.82 Vs 2.84, P=.021), but not for other symptoms/ We modified based on this theses results the known group validity under the analysis section, results and table 4  

Third, I suggest to discuss in a deeper way any limitations of the study.

Thank you for the valuable comment:

 We have now revised our limitations section as follows:

(1) this study was conducted in a tertiary care cancer center which may lead to selection bias and affect our study’s generalizability. (2) we only included patients with advanced cancer in this study; (3) a majority (96%) of participants were Jordanians in the study. We do not expect major differences in ESAS-A interpretation in other Arab countries; however, further testing is needed. (4) we did not conduct a test-retest reliability. (5) the responsiveness was only tested in the outpatient settings only, and thus these findings may be-not apply to hospitalized patients are typically acutely ill.

Round 2

Reviewer 1 Report

The abstract shows considerable improvement. There are a few typos and there that need to be corrected. For example, there is a set of unbalanced (  ). Check for these. 

I would appreciate a final review of the completed version once the edits are complete.

Author Response

Many thanks for your comments 

I have corrected all typos on the abstract based on your advise